# A Healthy Food Recommender System Using Collaborative Filtering and Transformers

## Abstract

Unhealthy eating habits are a major contributing factor to public health problems such as the globally rising obesity rate. One way to help solve this problem is by creating systems that can suggest better food choices in order to improve the way people eat. A critical challenge with these systems is making sure they offer 1) suggestions that match what users like, while also 2) recommending healthy foods. In this paper, we introduce a novel food recommender system that provides healthy food recommendations similar to what the user has previously eaten. We used collaborative filtering to generate recommendations and re-ranked the recommendations using a novel health score and a BERT embedding similarity score. We evaluated our system on human subjects by conducting A/B testing on several methods deployed in a web application.

## 1 Introduction

Unhealthy eating habits are a major concern worldwide, leading to various health issues such as obesity, heart disease, and diabetes Gracia (2020); Gonçalves et al. (2021). Numerous studies have highlighted the long-term effects of poor diet choices, emphasizing the urgent need for dietary improvements across populations Kandel (2023); Centers for Disease Control and Prevention (2023); Morris et al. (2016). As awareness about the impact of food on health grows, there has been a noticeable shift towards healthier eating habits. People are increasingly moving towards foods that not only satisfy their taste buds, but are also healthier.

In this context, recommendation systems have a critical role to play. By guiding users towards healthier food choices and showing the users the nutritious value of each food items, these systems can influence eating behaviors. Therefore, in this paper, we present an experiment in which we develop a system that matches food recommendations to a user's personal taste and also introduces healthier alternatives. This approach aims to facilitate a gradual shift in eating patterns, promoting a sustainable and healthy diet.

### 1.1 Motivation

In recent years, numerous food recommendation systems Gao et al. (2022); Meng et al. (2020); Pecune et al. (2020); Toledo et al. (2019); Zitouni et al. (2022) have been developed to predict individual preferences and guide food choices based on specific criteria. While these systems have shown relative success in understanding user preferences through historical interactions, there are still some improvements that can be done:

- Incorporating Nutritional Information: Many existing food recommender systems Gao et al. (2022); Toledo et al. (2019); Zitouni et al. (2022) do not integrate comprehensive nutritional data. The system proposed in this paper addresses this gap by integrating detailed nutritional profiles for each food item. It considers essential nutrients like protein, fiber, and vitamins while also taking into account harmful components such as sodium and sugars. This dual focus ensures that the recommendations not only cater to users' taste preferences, but also align with their overall health goals, thus avoiding the recommendation of unhealthy food items.
- Semantic Embeddings for Similar Foods: Previous systems have not fully leveraged the potential of semantic embeddings to capture the relationships between different food items.

By using BERT, our system can understand and identify foods with similar flavor profiles and culinary attributes. This allows for the recommendation of potentially healthier alternatives that are still similar to the user's preferred foods, thereby maintaining user satisfaction while promoting healthier eating habits.

- User Control and Customization: Our system allows users to customize their recommendations by specifying which nutrients to consider, assigning different weights to each nutrient, and choosing whether their recommendations should prioritize similarity, healthiness, or a balanced approach. This level of customization ensures that the recommendations are tailored to the user's specific dietary goals and preferences, providing a more personalized and interactive experience.

By addressing these limitations, this study aims to create a comprehensive and user-centric food recommender system that uses detailed nutritional analysis to offer personalized, health-oriented food recommendations. This approach not only bridges the gap between user preferences and nutritional science, but also promotes healthier dietary habits on a larger scale.

## 1.2 OBJECTIVES

The primary objective of this paper is to develop a recommender system that effectively aligns with individual dietary preferences while promoting healthier eating choices. This objective is underpinned by several key aims, which collectively address both the technological challenges and the practical applications of improving dietary habits through advanced computing techniques.

1. Personalization of Recommendations: The first aim is to personalize food recommendations to individual tastes and dietary requirements. This involves analyzing user data to understand preferences and eating patterns, which will then inform the system's suggestions. The system must be able to adapt to user feedback and evolve over time, ensuring that the recommendations remain relevant and appealing to each user.

2. Integration of Nutritional Guidelines: Another critical aim is to integrate comprehensive nutritional guidelines into the recommendation process. This involves not only suggesting foods based on taste compatibility, but also ensuring that these recommendations align with healthful eating practices.

3. Usability: The system should be user-friendly and accessible to a broad audience. This means designing an interface that is intuitive and engaging. While the focus of this paper is on the backend system, future work will study frontend UI/UX design and usability.

4. Evaluation: Finally, the system must include mechanisms for continuous evaluation and improvement. This involves setting up A/B testing to gather user responses and system performance data. These insights will be used to refine the algorithms and decide which models to incorporate in the deployed production system.

By achieving these objectives, this research aims to bridge the gap between nutritional science and consumer technology, creating a tool that not only makes healthier eating easier, but also more desirable. The ultimate goal is to contribute to public health outcomes by leveraging technology to influence and improve dietary behaviors on a large scale.

## 1.3 SYSTEM ARCHITECTURE

The architecture of the healthy food recommender system is designed to effectively integrate user preferences with nutritional data to generate personalized food suggestions that promote healthier eating habits. Figure 1 provides a visual overview of each component in the system, as detailed in the subsequent text.

1. **Generate Recommendations Using EASE/SVD**:

    The system initiates the recommendation process using one of two collaborative filtering methods: EASE (Embarrassingly Shallow Autoencoders) or SVD (Singular Value Decomposition). These algorithms are adept at processing sparse datasets and uncovering latent user-item interactions. A principal reason for integrating these methods is to encourage

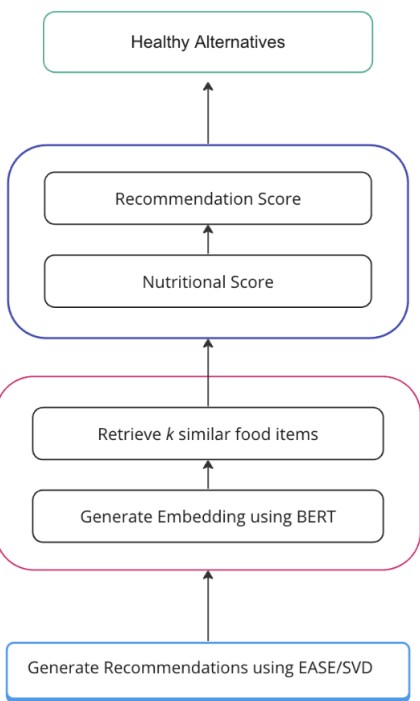

Figure 1: Architecture of the healthy food recommender system.

users to explore beyond their typical food choices, thereby expanding their dietary horizons and encouraging exploration outside their usual preferences.

2. **Generate Embedding Using BERT**:

   Subsequent to the initial recommendation phase, the system employs BERT to create contextual embeddings for each food item, which are essential for grasping the complex semantic relationships between items, allowing the system to suggest similar items.

3. **Retrieve k Similar Food Items**:

   Utilizing the embeddings generated by BERT, the system retrieves $k$ similar food items for each of the initially recommended items. While collaborative filtering with EASE/SVD enhances the diversity of options presented to the user, retrieving similar items with embeddings aligns these recommendations with the user's taste preferences.

4. **Nutritional Score**:

   Each food item undergoes an evaluation for its nutritional content to derive a health-centric score. This score incorporates various nutritional parameters to ensure that each recommendation supports health and wellness, helping prioritize foods that provide health benefits.

5. **Recommendation Score**:

   The system combines the similarity scores from BERT with the nutritional scores to calculate a comprehensive recommendation score for each item. This integration ensures that the final recommendations are both appealing to the user's taste and beneficial to their health.

6. **Healthy Alternatives**:

   The final output is a selection of healthy alternatives, chosen based on the highest recommendation scores. These alternatives are tailored to the user's preferences, yet offer healthier options, aiming to seamlessly integrate improved nutrition into their daily lives.

## 2 DATA & PREPROCESSING

To build our recommender system, we extracted training data from the meal descriptions provided by the COCO Nutrition Database Korpusik et al. (2014; 2016; 2019), which captured 41,424 users' food preferences. Using a convolutional neural network for semantic tagging Korpusik et al. (2017a); Korpusik & Glass (2017); Korpusik et al. (2017b); Korpusik & Glass (2018; 2019), we identified and extracted individual food items (both the natural language and the matching USDA food codes) that users had eaten. Our data pre-processing yielded a total of 11,548 different foods.

Next, we needed nutritional information for each food item to calculate health scores. For this, we used the USDA Food and Nutrient Database for Dietary Studies (FNDDS) U.S. Department of Agriculture, Agricultural Research Service (2023), which contains a comprehensive list of nutrients such as protein, saturated fats, dietary fiber, sodium, potassium, etc. We used the FNDDS as our food database, which we used to find similar and healthier alternatives to a given food item.

Since our recommendation algorithms, Singular Value Decomposition (SVD) Gower (2014) and Embarrassingly Shallow AutoEncoders (EASE) Steck (2019), rely on collaborative filtering techniques, creating a user-item interaction matrix was a necessary first step. We translated the food preferences data into a matrix where users were represented by rows, and food items were represented by columns.

To process this matrix, we experimented with two methods:

1. A count-based approach, recording how many times a user liked (i.e., ate) a specific food item. We normalized the data to minimize the influence of outliers using standardization. The z-score (normalized count) for a food $x$ is computed as:

$$z = \frac{(x - \mu)}{\sigma} \tag{1}$$

   where $\mu$ is the mean number of times each food was eaten, and $\sigma$ is the standard deviation. The normalized distribution of counts has a mean of 0 and a standard deviation of 1.

2. A binary approach, where the count is 1 if a user ate the food and 0 if not. The binary value for a user-item preference $b$ is:

$$b = \begin{cases} 1 & \text{if user likes (i.e., ate) the item} \\ 0 & \text{otherwise} \end{cases} \tag{2}$$

## 3 APPROACH

We used a two-step approach, in which a collaborative filtering algorithm generated the first set of 10 food recommendations based on the user's meal history, followed by a second step consisting of a novel health score and a BERT Devlin et al. (2018) embedding similarity score to retrieve 10 more healthy recommendations for each of the first 10.

### 3.1 STEP 1—COLLABORATIVE FILTERING FOR PRELIMINARY FOOD RECOMMENDATIONS

In our recommendation system, we investigated two collaborative filtering algorithms for the first pass at recommending foods: Singular Value Decomposition (SVD) Gower (2014) and Embarrassingly Shallow AutoEncoders (EASE) Steck (2019). Both models give personalized recommendations based on a user-item matrix.

#### 3.1.1 SVD MODEL

SVD is a common technique in collaborative filtering to generate recommendations. This method factors the user—item matrix into three matrices that uncover latent features underlying the interactions between users and items.

We have incorporated Stochastic Gradient Descent (SGD) Kiefer & Wolfowitz into our SVD model to optimize the prediction of how much a user would like a particular food item. This iterative

optimization method updates the model's parameters, gradually improving the accuracy of recommendations.

Moreover, to account for user and item biases—preferences or qualities affecting the ratings independently of each other—we introduced bias terms into our model. Specifically, we set both user and item biases to 0.01. Our loss function is defined as follows:

$$\min_{p,q,b} \sum_{u,i} \left(r_{ui} - \mu - b_u - b_i - p_u^T q_i\right)^2 + \lambda \left(\|p_u\|^2 + \|q_i\|^2 + b_u^2 + b_i^2\right) \tag{3}$$

We minimize the regularized squared error of the predicted ratings, where the predicted rating for a given user $u$ on item $i$, $r_{ui}$, is the sum of the overall average rating $\mu$, the user bias $b_u$, and the item bias $b_i$, adjusted by the interaction of the corresponding user and item latent factor vectors $p_u$ and $q_i$. Regularization is applied to prevent overfitting by penalizing the magnitude of the latent factors and biases, controlled by the regularization parameter $\lambda$.

## 3.2 EASE MODEL

To capture the co-occurrence and interaction strength between different items, we first computed the Gram matrix $\mathbf{G}$ from the given user-item matrix $\mathbf{X}$ as follows:

$$\mathbf{G} = \mathbf{X}^T \mathbf{X} \tag{4}$$

To prevent overfitting and to make the matrix invertible, we added a regularization parameter $\lambda$ to each of the diagonal elements of the Gram matrix as follows:

$$\mathbf{G}_{ii} = \mathbf{G}_{ii} + \lambda \tag{5}$$

An essential minimization step in the EASE model involves inverting the Gram matrix $\mathbf{G}$ to solve problems of the form $\mathbf{G}x = b$. This process can become computationally expensive as the number of users and items increases. To mitigate this, we employed matrix preconditioners, denoted as $\mathbf{P}$, which have demonstrated their efficiency in various applications to improve the matrix inversion process Bergamaschi et al. (2003); Mas et al. (2015); Colley et al. (2017). In this approach, the original problem $\mathbf{G}x = b$ is transformed into the preconditioned system $\mathbf{P}^{-1}\mathbf{G}x = \mathbf{P}^{-1}b$. Specifically, we used the Jacobi preconditioner, which leverages only the diagonal components of the Gram matrix, significantly reducing the computational overhead of matrix inversion. The Jacobi preconditioner $\mathbf{P}$ is computed as follows:

$$\mathbf{P} = \mathbf{D}(\mathbf{G}) \tag{6}$$

where the function $\mathbf{D}(\mathbf{X})$ constructs a diagonal matrix from the diagonal elements of $\mathbf{X}$. As the size and complexity of the data increase, more advanced solvers, such as the adaptive Algebraic Multigrid method Hu et al. (2019) and multilevel sparsifiers Hu & Lin (2024), can be employed to further enhance efficiency.

Our enhancements make the EASE model more efficient, allowing us to build the model twice as quickly as without the preconditioner, as shown in Table 1. We decided to use $\lambda = 0.01$ as it is the fastest.

Table 1: Latency (s) with and without Jacobi Preconditioner for different values of the regularization parameter ($\lambda$).

| REGULARIZATION ($\lambda$) | NO PRECONDITIONER (s) | JACOBI PRECONDITIONER (s) |
|---|---|---|
| 0.1 | 25.16 | 11.23 |
| 0.01 | 25.28 | 10.06 |
| 0.02 | 25.77 | 10.96 |
| 0.05 | 25.11 | 11.24 |

Table 2: Comparison of the top 5 most similar food items recommended by BERT and Word2Vec for the item *"Lucky Charms Cereal"*.

| METHOD | FOOD ITEM | SIMILARITY |
|--------|-----------|------------|
| **BERT** | Lucky Charms Cereal | **1.00** |
| | Chocolate Lucky Charms Cereal | 0.95 |
| | Chocolate Lucky Charms Cereal | 0.95 |
| | Lucky Charms Cereal Singlepak | 0.87 |
| | Lucky Charms | 0.86 |
| **Word2Vec** | Honey Nut Cheerios Cereal | **0.9999** |
| | Lucky Charms Cereal Singlepak | 0.9996 |
| | Cereal | 0.9957 |
| | Crispy Rice | 0.9931 |
| | Kix Cereal Bowlpak | 0.9928 |

### 3.3 STEP 2—ADDING HEALTHIER OPTIONS WITH A HEALTH SCORE AND SEMANTIC EMBEDDINGS

While the preliminary recommendations provided by the collaborative filtering step are similar to foods that the user has previously eaten, these recommendations are not necessarily healthier than what the user tends to eat, which is an important aspect of a food recommender system intended to improve diet. To address this shortcoming, we implemented a mechanism with a weighted score balancing similarity to what the user has previously eaten and a health score inspired by FDA guidelines. While we used equal weights for both parts, this weight could easily be adjusted by the user to either emphasize healthiness or likeability more.

Tables 2 and 3 show a comparison of food items based on their similarity to Lucky Charms Cereal, using two different methods: BERT and Word2Vec Mikolov et al. (2013). We chose to use BERT over Word2Vec as it looks at the whole sentence to understand the meaning of words given their context, whereas Word2Vec looks at words individually and uses fixed embeddings without context. Note that the second most similar food to Lucky Charms Cereal (after itself) using BERT is Chocolate Lucky Charms Cereal, whereas the most similar using Word2Vec is Honey Nut Cheerios Cereal, which is less similar semantically.

1. **Embedding Generation:** Each food item description is processed through the BERT model to generate vector embeddings. These embeddings capture the semantic content of each item in a high-dimensional space.

2. **Clustering Food Items:** We use the K-means clustering algorithm to group food items into clusters based on their semantic similarity. Each cluster comprises items that are contextually related.

3. **Finding Similar Items:**
   - **Cluster Selection:** We select the food item's cluster.
   - **Item Comparison:** Within the chosen cluster, we calculate the cosine similarity (see Table 2) between the embedding of the item and the embeddings of other items in its cluster.
   - **Top-k Retrieval:** Based on these similarity scores, we retrieve the top-*k* items that are most similar to the queried item, providing users with the most contextually related food alternatives.

#### 3.3.1 HEALTH SCORE

Calculating the health score of food items is an important process in our recommendation system. Our novel score helps us evaluate the nutritional value of each food item. To do this, we consider both healthy and unhealthy nutrients.

**Healthy Nutrient Contribution** We start by assessing the healthy nutrients in a food item. The contribution of these nutrients to the health score is calculated by comparing the amount present in the item to the recommended daily intake, known as the Daily Recommended Value (DV). Each nutrient is assigned a weight $W_h$ which signifies its importance to health. We multiply this weight by the percentage of the DV provided by the food item for that nutrient. Summing up these values for all healthy nutrients gives us the healthy contribution to the score, as follows:

$$\text{Healthy Contribution (HC)} = \sum_{h \in H} W_h \times \left( \frac{x_h}{DV_h} \right) \tag{7}$$

where:

- $H$ is a set of healthy nutrients.
- $W_h$ is a weight assigned to a nutrient $h$.
- $x_h$ is the amount of nutrient $h$ present in the food item.
- $DV_h$ is the Daily Recommended Value for nutrient $h$.

For healthy contribution, we consider the following nutrients and their respective weights—Protein (g): 1.2, Dietary Fibre (g): 1, Vitamin C (mg): 0.8, and Potassium (mg): 0.8.

**Unhealthy Nutrient Contribution** Similarly, we calculate the contribution of unhealthy nutrients. Instead of a direct comparison to the DV, we consider the proportion of these nutrients relative to an upper limit. We subtract this proportion from a base value (5% of the DV) considered low by the Food and Drug Administration (FDA) U.S. Food and Drug Administration (2023). We then multiply by the nutrient's weight. The sum of these values give us the unhealthy contribution:

$$\text{Unhealthy Contribution (UC)} = \sum_{u \in U} W_u \times \left( 0.05 - \frac{x_u}{DV_u} \right) \tag{8}$$

where:

- $U$ is a set of unhealthy nutrients with upper threshold values.
- $W_u$ is a weight for nutrient $u$.
- $x_u$ is the amount of nutrient $u$ in the food item.
- $DV_u$ is an upper intake level or threshold for nutrient $u$.

For unhealthy contribution, we consider the following nutrients and their respective weights—Sugar (g): 0.7, Sodium (mg): 0.7, and Total Saturated Fats (g): 1.

**Nutritional Score** The final health score for a food item is the sum of its healthy and unhealthy contributions, normalized to be between -1 and 1:

$$\text{Nutritional Score} = \text{HC} + \text{UC} \tag{9}$$

This score give us the overall nutritional quality of a food item, considering both its positive and negative attributes. To enhance the adaptability, it is possible to include additional nutrients specified by the user, such as Calcium, Magnesium, Carbohydrates, etc. in our nutritional score calculation. Furthermore, the weight $W$ assigned to each nutrient can be customized based on individual user preferences or specific dietary needs. This flexibility ensures that our nutritional assessments are both relevant and personalized.

### 3.3.2 RECOMMENDATION SCORE

After establishing the health score for our food items, we proceed to the final part of our recommendation system, which is the calculation of the Recommendation Score Rostami et al. (2020). This score encapsulates both the healthiness and the personal preferences (i.e., semantic similarity) of our recommendations.

Table 3: Three healthiest and unhealthiest food items based on the Nutritional Score.

| CATEGORY | FOOD ITEM | NUTRITIONAL SCORE |
|---|---|---|
| **Healthy Food Items** | Lemon Tea Drink Mix | **0.971** |
| | Low Calorie Tea Drink Mix Lemon Tea | 0.971 |
| | Great Value, Drink Mix, Apple | 0.96 |
| **Unhealthy Food Items** | Green Sugar | **-1.00** |
| | Belgian Milk Choc w/ Maple Sugar & Rice Crisps | -0.99 |
| | Holiday Candy Corn | -0.98 |

To determine the final Recommendation Score, we use the Nutritional Score with the Similarity Score, which reflects how closely a food item aligns with the user's taste preference. The formula for the Recommendation Score is given by:

$$R = \alpha \times \text{Similarity Score} + (1 - \alpha) \times \text{Nutritional Score} \qquad (10)$$

where $R$ is the Recommendation of a food item, $\alpha$ is a parameter that balances the importance of the similarity and health scores. By adjusting $\alpha$, we can skew the recommendations toward favoring either the similarity or the health aspect. In our implementation, we have chosen $\alpha = 0.5$, providing equal weight to both the health score and the similarity score.

## 4 EXPERIMENTS

### 4.1 AUTOMATIC EVALUATION

The Root Mean Square Error (RMSE), which we used to measure the accuracy of our predictions, is given by the equation:

$$RMSE = \sqrt{\frac{\sum_{ui}(r_{ui} - \hat{r}_{ui})^2}{|R|}} \qquad (11)$$

where $r_{ui}$ is the ground truth rating from user $u$ for item $i$ and $|R|$ is the number of ratings in our dataset.

We found collaborative filtering effective, even on our user-item matrix with a sparsity of 95.3%. Using SVD, the binary method for representing the user-item matrix was preferred as it resulted in a lower Root Mean Square Error (RMSE) of 0.68 on the validation data (30% of the data), compared to 0.76 with the z-score normalizer. Therefore, we used the binary normalizer to construct our final user-item matrix.

### 4.2 A/B TESTING WITH HUMANS

We developed a web application using Flutter to facilitate user interaction with our recommendation system (see Figure 2). New users are asked to select eight food items from a diverse set of 100 food items upon signing up. This initial selection serves to avoid the cold-start problem by providing initial data on user preferences.

Each user's selections were used as input to generate 10 recommendations using either EASE or SVD. Subsequently, for each of these 10 foods, the BERT similarity and health scores were used to generate a refined list of 10 healthier alternatives, from which users selected which recommendations they liked.

To evaluate the performance of the EASE and SVD models, we conducted A/B testing. The effectiveness of each model was measured by the ratio of liked recommendations to the total number of suggested items (i.e., 100). This approach helped us determine which model was more successful at aligning user preferences with healthier eating habits. Both models were tested on a total of 22 users to evaluate which garnered more likes for each user. As shown in Table 4, EASE achieved a 32%

Table 4: Human Evaluation of EASE and SVD Models.

| MODEL | AVERAGE LIKED PERCENTAGE |
| --- | --- |
| **EASE** | **0.32** |
| SVD | 0.26 |

average liking rate per user for recommended food items, while the SVD model recorded a slightly lower average of 26%. This data indicates that the EASE model was more effective in matching users with food items they preferred.

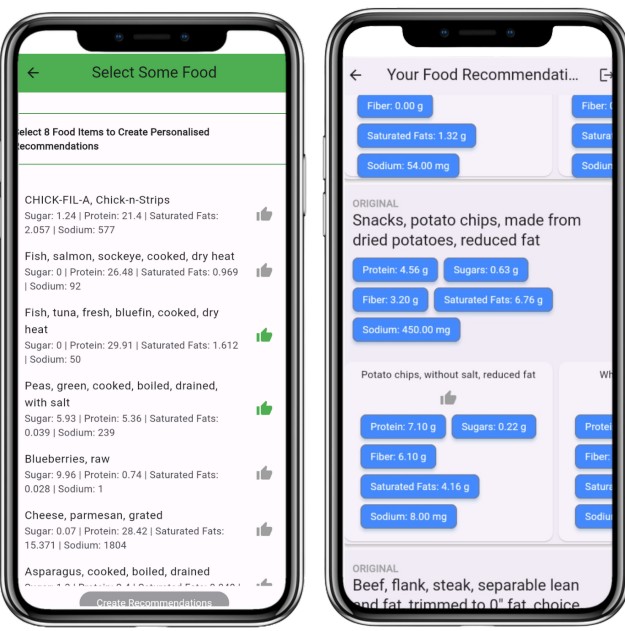

Figure 2: The Flutter web app for A/B testing. On the left, the user selects eight food items they like out of 100 options. On the right, the user selects which recommendations they like.

## 5 FUTURE WORK

The goal for future work is to deploy our recommender system within a publicly available free iOS application in the Apple Store to provide a seamless user experience for adopting healthier eating habits through personalized recommendations. Additionally, we plan to conduct more extensive A/B testing by using Amazon Mechanical Turk to gather feedback and preferences from a larger user base, which will allow us to validate our models' performance and refine our health score calculations across different demographics and dietary needs.

## 6 ETHICS

This research was conducted following approval from the University's Institutional Review Board (IRB). The study involved human subjects, and the potential benefits for participants included the discovery of new and healthy food options tailored to their personal preferences and nutritional needs, which could contribute to improved dietary habits and better overall health outcomes. Additionally, the participants had the opportunity to contribute to the development of a technology aimed at making healthy eating more accessible and personalized. We recognize the importance of safeguarding participants' privacy. All personal information, including dietary preferences and health

goals, was anonymized to protect individuals' identities. The data collected was securely stored and accessed only by the research team for the purpose of this study. We ensured that the information would not be used for any other purposes outside of the research, maintaining strict confidentiality throughout the study. There were no significant risks anticipated for participants, and any privacy concerns were addressed through these measures.

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
