# OpenReview forum: "A Healthy Food Recommender System Using Collaborative Filtering and Transformers"
_ICLR.cc/2025/Conference — Submitted to ICLR 2025_

### Official Review · Reviewer_dMdZ · 2024-10-31

**Soundness:** 1
**Presentation:** 2
**Contribution:** 1
**Rating:** 3
**Confidence:** 5

**Summary:**

The paper proposes a food recommender system that goes beyond traditional recommendation models by integrating nutritional information, leveraging semantic embeddings, and providing extensive customization options.This approach incorporates detailed nutrient data, including both beneficial (e.g., protein, vitamins) and potentially harmful components (e.g., sodium, sugars), ensuring recommendations align with users' health goals. It also uses BERT-based semantic embeddings to identify similar foods, allowing the system to suggest healthier alternatives without sacrificing user satisfaction. Moreover, the system allows users to customize their dietary priorities, such as favoring healthiness or flavor similarity, making the experience more interactive and tailored to individual goals. The main objectives include enhancing personalization, aligning recommendations with nutritional guidelines, ensuring usability, and establishing mechanisms for ongoing evaluation to refine and improve the recommendation model. Through this multifaceted approach, the system aims to promote healthier eating habits while maintaining high user satisfaction.

**Strengths:**

The paper attempts to integrate nutritional information with user preferences in food recommendations, which is a relevant and timely area of research. By using BERT for semantic embeddings, the authors leverage recent advancements in natural language processing to enhance the understanding of food relationships. This approach allows for a more nuanced recommendation that considers both taste and health aspects, potentially bridging a gap between traditional recommendation systems and health-conscious eating.

The methodology presented is well-structured, demonstrating a thoughtful design that combines established collaborative filtering techniques (EASE/SVD) with modern embedding approaches (BERT). The systematic evaluation of nutritional scores adds a robust layer to the recommendation process, ensuring that suggestions are not only appealing but also health-oriented.

The paper is clearly organized, with each section building logically on the previous one. The visual representation in Figure 1 provides a comprehensive overview of the system architecture, making it easier for readers to grasp the flow and interaction of components. The explanations of each step in the recommendation process are thorough, enhancing the reader's understanding of how user preferences and nutritional data are integrated.

The significance of this work lies in its potential impact on promoting healthier eating habits through personalized recommendations. By addressing the dual goals of satisfying user preferences and supporting health-related dietary choices, the system aligns with current trends in nutrition and wellness.

**Weaknesses:**

While the paper presents a food recommender system that integrates user preferences with nutritional data, it falls short in several key areas that limit its contribution to the field. The concepts largely rehash existing methods in food recommendation systems without introducing sufficiently novel insights. Although the authors attempt to integrate collaborative filtering and contextual embeddings, both EASE/SVD and BERT-based embeddings are well-established techniques. This combination, while functional, does not significantly advance the field or offer a new perspective on food recommendation challenges.

To make this work more impactful, I suggest the authors consider recent innovative approaches in food recommendation that extend beyond collaborative filtering and embeddings. For example, multi-objective optimization frameworks, such as those used by Gao et al. (2022), optimize for both health goals and user satisfaction simultaneously. Additionally, reinforcement learning-based recommender systems that adjust recommendations based on real-time user feedback could provide a more adaptive approach than static collaborative filtering models. Moreover, incorporating multi-view learning or multi-modal embeddings (e.g., combining user activity data, time of day, and food availability) could help capture a richer, context-aware recommendation space. These techniques, documented in recent studies like Zitouni et al. (2022), might allow the system to adapt recommendations dynamically, responding to factors that influence dietary choices.

Further innovation could be achieved through novel adaptations of existing methods. For example, graph-based models that map both food similarity and nutritional content in a unified framework could provide a complex, relational approach, setting this work apart from traditional collaborative filtering. These alternative formulations would help the authors demonstrate stronger novelty, particularly if they compared the proposed system against these advanced approaches. By exploring or incorporating some of these techniques, the paper could provide a more compelling case for its system’s uniqueness and address the observed gaps in novelty. This would help strengthen its contribution to the development of food recommender systems that support healthier dietary choices.

The evaluation does not incorporate essential metrics such as Hit Rate, which is particularly important for food recommendation systems because it directly measures the system's success in presenting items the user is likely to engage with. For a system focused on promoting healthier food choices, accurately predicting relevant items is crucial to encouraging adoption and sustained user interest. Including Hit Rate as an evaluation metric would provide insight into the recommendation system's accuracy and appeal. Additionally, an ablation study is needed to assess the contributions of each component of the system, helping to clarify the impact of the collaborative filtering methods, BERT embeddings, and nutritional scoring on the overall performance. This would provide a clearer understanding of the strengths and weaknesses of the proposed approach. While the paper is structured and contains clear descriptions of the components involved, this clarity does not compensate for the lack of originality and rigorous evaluation. The overall presentation does not highlight any unique or insightful findings that would warrant further exploration or application.

The paper fails to address a substantial gap in the literature or offer a new solution that could influence the domain of food recommendations. Its contributions seem incremental rather than transformative, and the proposed system does not advance our understanding of dietary recommendation systems in a meaningful way. In conclusion, the paper does not sufficiently present novel contributions or robust evaluations that would warrant acceptance.

**Questions:**

1. Novelty of Contribution:
   - Question: Can you clarify how your approach offers a novel contribution to the field of food recommendation systems? Specifically, what unique problem formulations or innovative adaptations to existing methods do you believe set your work apart from prior studies?

2. Dataset Limitations:
   - Question: Your current evaluation is based on a single dataset. How do you ensure that your findings are generalizable across different user demographics and food preferences?
   - Suggestion: Consider expanding your evaluation to include additional datasets that vary in user behavior and food types. This would strengthen the robustness of your results.

3. Model Evaluation Metrics:
   - Question: Why did you choose to exclude certain evaluation metrics such as Hit Rate? How do you plan to address this gap in your evaluation?
   - Suggestion: Incorporating Hit Rate and other relevant metrics into your evaluation would provide a clearer picture of your system's performance. Consider revising your evaluation framework to include these metrics.

4. Ablation Study:
   - Question: Have you considered conducting an ablation study to assess the individual contributions of different components of your recommendation system?
   - Suggestion: An ablation study would be valuable for demonstrating the effectiveness of each part of your system, such as the collaborative filtering methods and BERT embeddings. This could clarify how each component contributes to the overall performance of your recommendations.

5. User Preferences Integration:
   - Question: How does your system adapt to changing user preferences over time? What mechanisms are in place to incorporate user feedback into the recommendation process?

6. User Interface Considerations:
   - Question: What plans do you have for user interface and user experience (UI/UX) design? How will these considerations impact user engagement and satisfaction with the recommendations?

---

### Official Review · Reviewer_cxXM · 2024-11-01

**Soundness:** 2
**Presentation:** 3
**Contribution:** 2
**Rating:** 3
**Confidence:** 4

**Summary:**

The authors present a novel approach to providing food recommendations that are both healthy and tasty for customers. The proposed solution combines the power of Collaborative Filtering (CF) with a semantic-based similarity score for the food recommended in the first stage (CF). This, combined with a final health score, should give users the possibility to explore new food alternatives that they enjoy but which are also healthy.

The solution is straightforward: SVD can be used in the first stage, followed by a BERT-based similarity score for ingredients and food nutrition in the second stage, alongside a novel health score.

The idea is interesting, and the authors also conducted an A/B test. The main issue here is the lack of comparisons with previous works, for instance (but not just limited to) Pecan (2020). I strongly suggest the authors consider ways to perform quantitative comparisons with experiments (not online experiments, offline with a dataset would be ok), ideally using the same datasets as prior studies where possible. While a qualitative assessment was included, a clear quantitative comparison is needed to understand the benefits of the proposed solution.

Additionally, further insights are needed: which recommendations were effective, which were not, and why? How can these be improved? An A/B test that only compares different CF methods does not sufficiently add value to the assessment of the work.

**Strengths:**

- An impactful problem, they are working on healthy food recommendations
- Straightforward solution, simple and well described. The paper is easy to follow

**Weaknesses:**

- No comparisons. They have to compare previous works w.r.t. proposed solutions, a quantitative comparisons would help to understand the real benefits of this solution
- No baselines, really the weakness is around the fact that the authors should spend sometime trying to compare their solution

**Questions:**

Was there any particular reasons you have decided to not compare your solution w.r.t previous works?
Are you able to use previously used, proposed datasets?
Why it was important to check the difference in performance for SVD vs EASE?

---

### Official Review · Reviewer_rttq · 2024-11-05

**Soundness:** 2
**Presentation:** 2
**Contribution:** 1
**Rating:** 1
**Confidence:** 4

**Summary:**

The paper presents a novel approach to recommend food products that could, at the same time, meet two constraints: 1) suggest accurate items according to which users like; 2) suggest healthy food. Unlike previous similar approaches, the authors propose to incorporate nutritional information within the recommendation pipeline, such as essential nutrients and possibly harmful components. Then, they also propose to adopt BERT to generate semantic embeddings to learn the relationships between different food items. Finally, the system allows to customize the user experience by letting the user decide which nutrients to take into account in the received recommendations. The overall approach consists of the following steps. First, the authors exploit two popular collaborative filtering algorithms, namely, EASE and SVD, to generate traditional food item recommendations. Second, to address health, the framework generates and injects vector embeddings (through BERT) that are used to cluster food products and find the top-k similar list of items to the ones previously recommended; this allows the user to receive alternatives. After that, the nutritional score is computed with the healthy and unhealthy nutrition contributions. Finally, the nutritional score and the similarity score (from the clustering) are combined to obtain a final recommendation score. Results of the offline evaluation showed that the SVD method could outperform EASE. In addition to this, the authors conducted online A/B testing with web applications, which showed that the EASE method was actually outperforming the SVD one.

**Strengths:**

\+ The paper tackles an interesting aspect of food recommendation, namely, the introduction of healthy suggestions alongside the usual accurate ones

\+ The authors conducted both an offline and an online evaluation of the proposed framework

**Weaknesses:**

\- Despite the detailed introduction and presentation of the motivations behind the work, it gets very difficult to tell how the proposed approach is different from the related literature in terms of rationales, concepts, and technical aspects

\- Some important related work is missing from the paper, here to mention a few: [i, ii, iii, iv, v, vi, vii, viii]

\- Some technical details of the proposed framework are not adequately justified (see the questions section)

\- The experimental part of the paper seems quite limited and very few evaluation dimensions are taken into account; in the offline evaluation, I would have added comparisons against some of the missing citations mentioned in the second weaknesses point

**References**

[i] Zhenfeng Lei, Anwar Ul Haq, Adnan Zeb, Md Suzauddola, Defu Zhang: Is the suggested food your desired?: Multi-modal recipe recommendation with demand-based knowledge graph. Expert Syst. Appl. 186: 115708 (2021)

[ii] Weiqing Min, Shuqiang Jiang, Ramesh C. Jain: Food Recommendation: Framework, Existing Solutions, and Challenges. IEEE Trans. Multim. 22(10): 2659-2671 (2020)

[iii] Wenjie Wang, Ling-Yu Duan, Hao Jiang, Peiguang Jing, Xuemeng Song, Liqiang Nie: Market2Dish: Health-aware Food Recommendation. ACM Trans. Multim. Comput. Commun. Appl. 17(1): 33:1-33:19 (2021)

[iv] Alain D. Starke, Cataldo Musto, Amon Rapp, Giovanni Semeraro, Christoph Trattner: "Tell Me Why": using natural language justifications in a recipe recommender system to support healthier food choices. User Model. User Adapt. Interact. 34(2): 407-440 (2024)

[v] Alessandro Petruzzelli, Cataldo Musto, Michele Ciro Di Carlo, Giovanni Tempesta, Giovanni Semeraro: Recommending Healthy and Sustainable Meals exploiting Food Retrieval and Large Language Models. RecSys 2024: 1057-1061

[vi] Mehrdad Rostami, Vahid Farrahi, Sajad Ahmadian, Seyed Mohammad Jafar Jalali, Mourad Oussalah: A novel healthy and time-aware food recommender system using attributed community detection. Expert Syst. Appl. 221: 119719 (2023)

[vii] Alain Starke, Ayoub El Majjodi, Christoph Trattner: Boosting Health? Examining the Role of Nutrition Labels and Preference Elicitation Methods in Food Recommendation. IntRS@RecSys 2022: 67-84

[viii] Alain D. Starke, Christoph Trattner: Promoting Healthy Food Choices Online: A Case for Multi-List Recommender Systems. IUI Workshops 2021

**Questions:**

I have some questions, especially regarding the novelty of the work and its technical aspects:

1) How is the presented work different and novel from some of the papers that were not mentioned (refer, especially, to [vii-viii] from the references paragraph in the weaknesses section)?

2) Why do the authors adopt SVD and EASE as base recommender models? Is that only because they are well-established methods in the literature, or there is a reason connected to the specific domain (i.e., food recommendation)?

---

### Official Review · Reviewer_RqKP · 2024-11-06

**Soundness:** 2
**Presentation:** 2
**Contribution:** 1
**Rating:** 1
**Confidence:** 5

**Summary:**

This paper addresses the food recommendation issues by considering food nutrition beyond collaborative preferences. The collaborative preferences are captured by two Collaborative Filtering methods (SVD and EASE). At the same time, the food nutrition scores are computed via two authors-defined empirical equations (Healthy Contribution and Unhealthy Contribution). Experiments are conducted on a filtered COCO Nutrition Database with 11.5K foods. A human evaluation with 22 users is also conducted to compare the two SVD and EASE models.

**Strengths:**

S1: It is nice to consider the characteristics of food items, i.e., the food nutrition, when recommending the food items to users.

**Weaknesses:**

W1: The core of healthy nutrients and unhealthy nutrients (Equations 7-8) is not well motivated. Why the four kinds of nutrition (Protein, Dietary Fibre, Vitamin, Potassium) are defined as healthy while the other three (Sugar, Sodium, Saturated Fats) are unhealthy?

W2: The case of healthy and unhealthy is much varied with different genders and ages (old, young). The universe definition in Equations 7-8 might be inappropriate, especially the fixed weights.

W3: The evaluation metrics on recommendation should include ranking metrics like AUC, NDCG, and Recall. The RMSE metric only is not enough in Section 4.1 AUTOMATIC EVALUATION.

W4: Key details are missing in evaluation and implementation. What are the statistics of the datasets, how many users, how many interactions, how to split the training set and test set, how many words and tokens are in the food item descriptions, and how long of the average length of the food item descriptions, what is the K in K-means, any visualization e.g. t-SNE on the clustering foods?

**Questions:**

Comments

C1: Just like we use the RMSE metric to measure the recommendation results, do we have some kind of “healthy metrics” to evaluate the recommendation results?

C2: In Table 4, the recommendation performance is only 0.32, can this result be deployed in real web applications?

C3: On Line 079, it says “The system must be able to adapt to user feedback and evolve over time”, do the experiments investigate this “evolve over time” issue?

C4: On Line 168, it says “extracted individual food items (both the natural language and the matching USDA food codes)”, then what is the accuracy of this extraction, especially the “matching” accuracy on the USDA food codes?

---

### Meta-Review · Area_Chair_K4mg · 2024-12-19

**Metareview:**

This study focuses on how to integrate nutrition health information into food recommendation systems to provide better recommendation results. However, as pointed out by the reviewers, the paper has significant weaknesses in research motivation, experimental setup, and comparison with SOTA models. These issues result in limited innovation of the proposed method and make the current experimental comparisons insufficient to validate the method's effectiveness. Furthermore, the authors do not reply to reviewers' concerns, and all reviewers agree that this paper is not ready by far. Therefore, my meta-review recommendation is to reject.

**Additional Comments On Reviewer Discussion:**

The authors do not reply to reviewers and all reviewers agree this paper is not ready.

---

### Decision · Program_Chairs · 2025-01-22

Reject